# Quality Assessment of Global Ocean Island Datasets

**Yijun Chen [1], Shenxin Zhao [1]**[ID]**, Lihua Zhang [2] and Qi Zhou [1,3,*]**[ID]

[1] School of Geography and Information Engineering, China University of Geosciences, Wuhan 430074, China
[2] Department of Military Oceanography and Hydrography & Cartography, Dalian Naval Academy, Dalian 116018, China
[3] International Research Center of Big Data for Sustainable Development Goals, Beijing 100094, China
[*] Correspondence: zhouqi@cug.edu.cn; Tel.: +86-151-7238-2436

**Abstract:** Ocean Island data are essential to the conservation and management of islands and coastal ecosystems, and have also been adopted by the United Nations as a sustainable development goal (SDG 14). Currently, two categories of island datasets, i.e., global shoreline vector (GSV) and OpenStreetMap (OSM), are freely available on a global scale. However, few studies have focused on accessing and comparing the data quality of these two datasets, which is the main purpose of our study. Specifically, these two datasets were accessed using four $100 \times 100$ (km$^2$) study areas, in terms of three aspects of measures, i.e., accuracy (including overall accuracy (OA), precision, recall and F1), completeness (including area completeness and count completeness) and shape complexity. The results showed that: (1) Both the two datasets perform well in terms of the OA (98% or above) and F1 (0.9 or above); the OSM dataset performs better in terms of precision, but the GSV dataset performs better in terms of recall. (2) The area completeness is almost 100%, but the count completeness is much higher than 100%, indicating the total areas of the two datasets are almost the same, but there are many more islands in the OSM dataset. (3) In most cases, the fractal dimension of the OSM dataset is relatively larger than the GSV dataset in terms of the shape complexity, indicating that the OSM dataset has more detail in terms of the island boundary or coastline. We concluded that both of the datasets (GSV and OSM) are effective for island mapping, but the OSM dataset can identify more small islands and has more detail.

**Keywords:** SDG 14; global shoreline vector; OpenStreetMap; data quality; accuracy; completeness; fractal dimension

## 1. Introduction

Ocean islands, which are defined as lands that are entirely surrounded by ocean waters, are not only homes to many unique plants and animals around the world, but also living places for human beings. It has been estimated that approximately 550 million people, 9–10% of the world's population, live on islands [1]. Currently, the conservation and management of islands and coastal ecosystems are receiving significant attention [2] because islands are now threatened by rising sea levels (caused by climate change [3–5]), natural disasters (such as storms, tsunamis, and volcanic eruptions [6,7]), and human activity (such as overfishing and island degradation [8–10]). In order to deal with these challenges, the management and protection of marine and coastal ecosystems has been adopted by the United Nations as one of the 17 sustainable development goals (SDGs), specifically SDG 14: Conserve and sustainably use the oceans, seas, and marine resources for sustainable development [11]. Currently, available and large-scale geospatial data related to islands are especially needed for the evaluation and monitoring of various indicators related to SDG 14.

### 1.1. Related Works

Remote sensing has been viewed as a potential technology for detecting islands and relevant characteristics, such as temperature and land-use change. Dong et al. [12] developed

a simple method for mapping the inundation frequency of coral reefs in the Spratly Islands in the South China Sea using time series Landsat-8 OLI images. Immordino et al. [13] used Sentinel-2 multispectral data to map different types of habitats, including corals, seagrasses, and mangroves, in the Palau Republic in the Pacific Ocean. Lyons et al. [14] presented a framework capable of mapping coral reef habitats from individual reefs to entire barrier reef systems and across vast ocean extents, using high-resolution remote sensing data available on a global scale. Zhuang et al. [15] proposed a technical framework for automatic coral reef extraction based on an image filtering strategy and spatio-temporal similarity measurements of pixel-level Sentinel-2 image time series. Mikelsons et al. [16] developed a methodology to derive a global medium resolution (250 m) land mask or water mask from several existing data sources. In terms of island characteristics, Král and Pavliš [17] produced the first detailed land-cover map of Socotra Island using Landsat 7 ETM+ data. Révillion et al. [18] developed a land-use/land-cover product based on remote sensing processing of high spatial resolution satellite images acquired by the SPOT 5 satellite between December 2012 and July 2014. Chen et al. [19] used Landsat data for eight periods from 1984 to 2020 to explore the spatial and temporal characteristics of the land-use landscape pattern of Zhoushan Island, China. Holdaway et al. [20] analyzed changes in the land area on 221 atolls (ring-shaped coral islands or reefs) in the Indian and Pacific Oceans. Leihy et al. [21] applied a spatial-temporal gap-filling method to high-resolution (~1 km) land surface temperature observations for 20 Southern Ocean islands.

Although extensive studies have been conducted to detect islands and the relevant characteristics of islands, most have focused on proposing approaches, methods or technical frameworks, rather than producing available island data for public use. To address this gap, Sayre et al. [22] recently developed a 30 m spatial resolution global shoreline vector (GSV) from annual composites of 2014 Landsat 7 satellite images. The GSV dataset has three classes of islands: continental mainlands, islands greater than 1 km$^2$ and islands smaller than 1 km$^2$. More importantly, this dataset was not only made available globally but also open to the public. As another alternative, the OpenStreetMap (OSM) data, edited by global volunteers, can also be used for acquiring geospatial data related to islands. There are several benefits of using the OSM data [23]. First, it is being edited by global volunteers and thus has a global coverage. Second, the OSM data can also be freely acquired for public use. Third, the data contains many map features (e.g., roads, buildings and land-uses); more importantly, islands (https://wiki.openstreetmap.org/wiki/Tag:place%3Disland, accessed on 27 April 2021) and islets (https://wiki.openstreetmap.org/wiki/Tag:place%3Dislet, accessed on 27 April 2021) data can also be acquired directly from OSM.

Despite these available island datasets (GSV and OSM), to the best of our knowledge, few studies have paid attention to the data quality of these datasets. The GSV dataset has only been validated using visual inspection rather than quantitative assessment [22]. Many concerns have also been raised about the data quality of OSM because the data was edited by global volunteers from different countries [24], and of different ages and educational backgrounds [25]. Although extensive studies have been conducted to assess OSM data quality in terms of roads [26–28], buildings [29–31], land-cover, and land-uses [32–34], there is still a lack of research assessing OSM data quality in terms of islands and/or islets.

### 1.2. Aim and Contributions

Therefore, the purpose of our study is to assess and compare the data quality of two existing island datasets (GSV and OSM). Moreover, this study has two main contributions.

(1) Different measures (including accuracy, completeness and shape complexity) were designed for assessing the data quality of island datasets.
(2) Both the GSV and OSM datasets were not only assessed but also compared, in order to investigate which performed the best.

### *1.3. Organization*

The paper is structured as follows: Section 2 describes the study area and data. Section 3 presents the designed measures that were used to assess and compare GSV and OSM island datasets. Section 4 reports the results and analyses. Sections 5 and 6 comprise the discussion and conclusion, respectively.

## 2. Study Area and Data

### *2.1. Study Area*

Four $100 \times 100$ km$^2$ regions were chosen as the study areas (as shown in Figure 1). These regions were selected for several reasons: First, they are located in different geographical regions of the world, namely the Atlantic Ocean, the Arctic Ocean, the Indian Ocean and the Pacific Ocean (see Appendix A). Second, the size and pattern of the islands vary between the different regions, as indicated in Table 1. For instance, the islands in study area II are relatively large, while those in study area III are much smaller. In contrast, the islands in study areas I and IV show a combination of different sizes. Third, and most important, four different study areas were chosen to minimize any potential bias in the analysis.

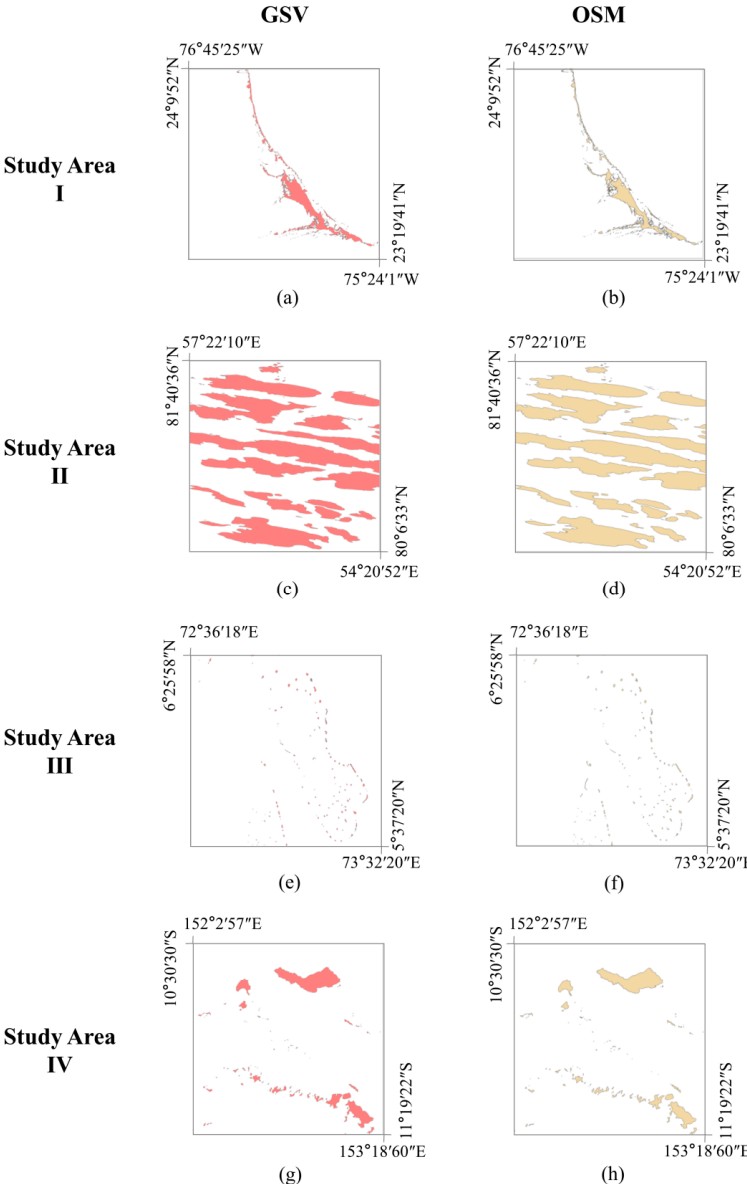

**Figure 1.** The GSV and OSM datasets of four study areas: I (**a**,**b**); II (**c**,**d**); III (**e**,**f**); and IV (**g**,**h**).

**Table 1.** The statistics of islands in the four study areas.

| Study Area | Geographical Region | Total Number of Islands | | Average Size of Islands (m$^2$) | |
|:---:|:---:|:---:|:---:|:---:|:---:|
| | | GSV | OSM | GSV | OSM |
| I | Atlantic Ocean | 417 | 764 | $6.9 \times 10^5$ | $3.4 \times 10^5$ |
| II | Arctic Ocean | 56 | 73 | $5.9 \times 10^7$ | $4.5 \times 10^7$ |
| III | Indian Ocean | 151 | 172 | $2.5 \times 10^5$ | $2.4 \times 10^5$ |
| IV | Pacific Ocean | 115 | 136 | $3.7 \times 10^6$ | $3.0 \times 10^7$ |

### 2.2. Data

Two categories of open island datasets (global shoreline vector and OpenStreetMap) were used for the analysis (Table 2).

**Table 2.** The attributes of the two island datasets (GSV and OSM).

| Dataset | Type/Tag | Definition |
|:---:|:---:|:---:|
| Global Shoreline Vector (GSV) | Continental mainlands | Northern America, Southern America, Africa, Australia, Eurasia |
| | Large islands | Islands that are larger than 1 km$^2$ |
| | Small islands | Islands that are smaller than 1 km$^2$ |
| OpenStreetMap (OSM) | natural = coastline | The mean high water springs line along the coastline at the edge of the sea |
| | place = island | Any piece of land that is completely surrounded by water and isolated from other significant landmasses |
| | place = islet | Any very small island |

- Global shoreline vector (GSV): The dataset is a 30-m spatial resolution global shoreline vector, which was produced based on annual composites of 2014 Landsat satellite images [22]. This dataset includes 340,691 islands in total, divided into three classes, i.e., 5 continental mainlands, 21,818 big islands greater than 1 km$^2$ and 318,868 islands smaller than 1 km$^2$ (Table 2). The dataset was acquired on 27 April 2021 from the website https://rmgsc.cr.usgs.gov/gie.
- OpenStreetMap (OSM): The dataset were freely acquired from Planet OSM: https://planet.openstreetmap.org/ (accessed on 13 February 2023), acquired in 2021. The platform provides all OSM data on a global scale. Each object in OSM has at least one tag (consisting of a key and a value) to describe the attribute of this object. As an example, if an OSM object is tagged with "place (key) = islet (value)", it means that this object is a small island in the sea. Moreover, in our study, three different tags (i.e., natural = coastline, place = island and place = islet) relating to islands were extracted from OSM data (Table 2). In addition, the extracted data, originally saved in a pdf format, were converted into a shapefile format for the analysis, because the latter format can be processed by most geographic information system (GIS) software (e.g., ArcGIS and QGIS).

## 3. Methodology

The two island datasets (GSV and OSM) were evaluated based on three aspects: (1) accuracy, (2) completeness, and (3) boundary complexity. This is because accuracy and completeness are quality measures defined by ISO (International Organization for Standardization 2013 [35]) that are widely used to assess the quality of various types of geospatial data, such as roads, buildings, and land-cover/land use. Furthermore, geometry irregularity or complexity

is often analyzed when investigating coastlines [36]. Specifically, the workflow is shown in Figure 2, and the corresponding evaluation measures are introduced below.

**Figure 2.** The workflow of evaluation method.

### 3.1. Accuracy

Accuracy is used to measure whether the islands in each open dataset are represented correctly. As a reference island dataset is not freely available, the basis of our evaluation approach is to compare each open dataset with a set of sampling points that were visually interpreted from Google Earth. Specifically,

- First, a set of sampling points with an interval of 2 km was acquired from each study area, resulting in a total of 2500 sampling points for each study area.
- Next, the reference classification of each sampling point (either 'island' or 'non-island') was visually interpreted from the corresponding satellite image in Google Earth, which was taken around the year 2021.
- Subsequently, all sampling points for each study area were overlaid on each open dataset (e.g., GSV or OSM) to determine the predicted classification (either 'island' or 'non-island') of each sampling point. Specifically, if a sampling point was located within the polygon of an island, it was classified as 'island'; otherwise, it was classified as 'non-island'.
- Finally, the predicted classification of each sampling point was compared with the corresponding reference classification, using four different measures: overall accuracy (OA), precision, recall, and F1. These measures were chosen because they have been widely used to evaluate the performance of classification problems [37,38].

$$OA = \frac{TP + TN}{TP + FP + TN + FN} \times 100\% \tag{1}$$

$$Precision = \frac{FP}{FP + TN} \times 100\% \tag{2}$$

$$Recall = \frac{TP}{TP + FN} \times 100\% \tag{3}$$

$$F1 = \frac{2 \times Precision \times Recall}{Precision + Recall} \times 100\% \tag{4}$$

where $TP$ denotes the number of sampling points that were identified as 'island' in both the open dataset and Google Earth; $TN$ denotes the number of sampling points that were identified as 'non-island' in both the open dataset and Google Earth; $FP$ denotes the number of sampling points that were classified as 'island' in the open dataset but interpreted as 'non-island' in Google Earth; and $FN$ denotes the number of sampling points that were classified as 'non-island' in the open dataset but interpreted as 'island' in Google Earth.

### 3.2. Completeness

Completeness indicates how well a region has been mapped. As a freely available reference island dataset was not available, it was impossible to calculate the actual completeness. As an alternative, we compared the relative differences between the two island datasets (GSV and OSM). Specifically, two measures (area completeness and count completeness) were used to assess completeness [29]. The two measures (called $C_{area}$ and $C_{count}$) are described as follows.

$$C_{area} = \frac{A_{OSM}}{A_{GSV}} \times 100\% \tag{5}$$

$$C_{count} = \frac{N_{OSM}}{N_{GSV}} \times 100\% \tag{6}$$

where $A_{GSV}$ and $A_{OSM}$ denote the total areas of islands in the GSV and OSM datasets, respectively, and $N_{OSM}$ and $N_{GSV}$ denote the total number of islands in the OSM and GSV datasets, respectively.

Furthermore, in order to investigate how small an island can be identified using these datasets, we compared the number of islands in the two datasets (GSV and OSM) in terms of different area intervals (i.e., $0–10^2$, $10^2–30^2$, $30^2–50^2$, $50^2–100^2$, $100^2–1000^2$ and $>1000^2$ m$^2$). Additionally, we investigated whether the islands identified in these datasets actually exist or were incorrectly identified, which was achieved through visual interpretation using Google Earth.

### 3.3. Shape Complexity

The shape complexity denotes the complexity of an object's shape or boundary. For this study, we analyzed the shape complexity of islands in each dataset to investigate which dataset (i.e., GSV or OSM) has more details. Specifically, we employed the box-counting method which has been widely applied to analyze the shape or boundary complexity of coastlines, to calculate this measure [36,39].

The main steps for using the box-counting method are:

- First, the islands in each dataset (GSV or OSM), which were originally represented by polygons, were converted into lines (or boundaries).
- Then, the lines or boundaries in each dataset and each study area (i.e., I, II, III or IV) were respectively overlaid with regular grids of different sizes (i.e., 10, 30, 50, 100, 300, 500 and 1000 m, Figure 3). For each grid cell, not only was the size of the grid cell ($r$) recorded but also the number of grid cells that intersected with a line or boundary ($Nr$) was calculated.
- After that, the natural logarithms of each pair of $r$ and $N_r$ were calculated. That means each pair of $r$ and $N_r$ was converted into $In(r)$ and $In(N_r)$, respectively.
- Lastly, a linear function was used to fit multiple pairs of $In(r)$ and $In(N_r)$ of different grid sizes, that is,

$$In(N_r) = -Dln(r) + lnC \tag{7}$$

where $C$ denotes a constant; and $D$ denotes the fractal dimension. Commonly, the larger the fractal dimension, the more details of islands in a dataset.

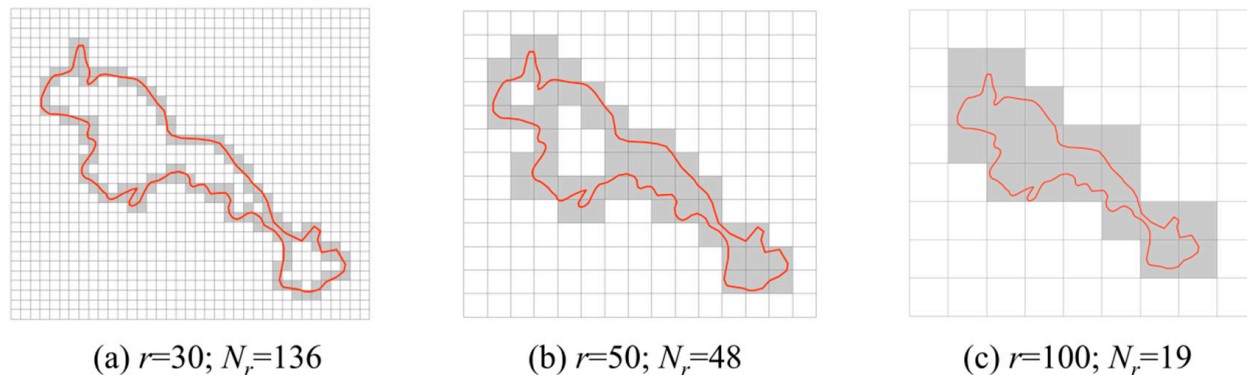

(a) *r*=30; *N*$_r$=136  (b) *r*=50; *N*$_r$=48  (c) *r*=100; *N*$_r$=19

**Figure 3.** The principle of the box-counting method, using different grid sizes, i.e., (**a**) 30 m; (**b**) 50 m; and (**c**) 100 m.

## 4. Results and Analyses

### 4.1. Results of Accuracy

First of all, Figure 4 shows the evaluation results of the two island datasets (GSV and OSM), in terms of accuracy. The specific values are listed in Appendix B.

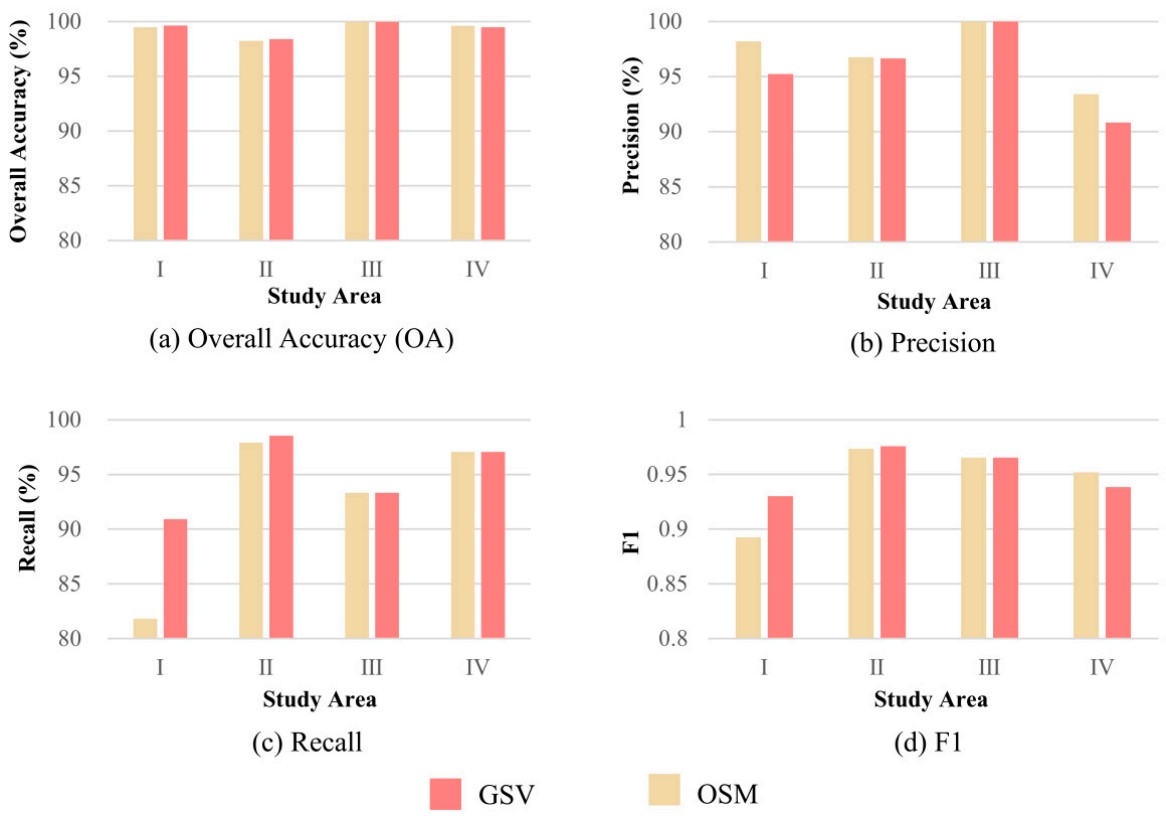

**Figure 4.** The evaluation results of two island datasets and four regions (I–IV), in terms of overall accuracy (**a**), precision (**b**), recall (**c**), and F1 (**d**).

We can see from Figure 4 and Appendix B that:

(1)    The overall accuracy (OA) is almost 100% for both datasets (GSV and OSM), and even the lowest OA value is higher than 98%. This indicates the effectiveness of using the two datasets for mapping islands. Moreover, for two out of the four study areas, the OA is slightly higher for the GSV dataset than for the OSM dataset. However, this is

the opposite case for study area IV, which suggests that performance may vary with different study areas.

(2) In most cases, precision is higher for the OSM dataset. Taking study area III as an example, precision is 93.40% for the OSM dataset, which is higher than that (90.83%) for the GSV dataset. Despite this, most precision values are higher than 90%, indicating that most sampling points identified as 'island' in these island datasets have also been classified as 'island' when referring to Google Earth.

(3) Unlike precision, recall values are higher for the GSV dataset than for the OSM dataset in most cases. Taking study area I as an example, the recall value is only 81.82% for the OSM dataset, which is much lower than that (90%) for the GSV dataset. Despite this, all recall values are higher than 90%, indicating that most sampling points classified as 'island' in Google Earth have also been identified as 'island' in these island datasets.

(4) The best performance of the two island datasets also varies with different study areas in terms of F1. Specifically, the GSV dataset performs better than the OSM dataset for study areas I and II, but this is the opposite case for study area IV.

Further, two examples are used to illustrate the performance of two island datasets (GSV and OSM) by overlapping them with satellite images in Google Earth (Figure 5). Figure 5a shows that the OSM dataset provides a more precise identification of the island than the GSV dataset. For instance, the yellow sampling point in Figure 5a was visually interpreted as 'non-island', but the GSV dataset identified it as 'island'. Figure 5b shows that the GSV dataset yields a more complete identification of the island compared to the OSM dataset. For instance, the yellow sampling point in Figure 5b was visually interpreted as 'island' in Google Earth, but the OSM dataset identified it as 'non-island'.

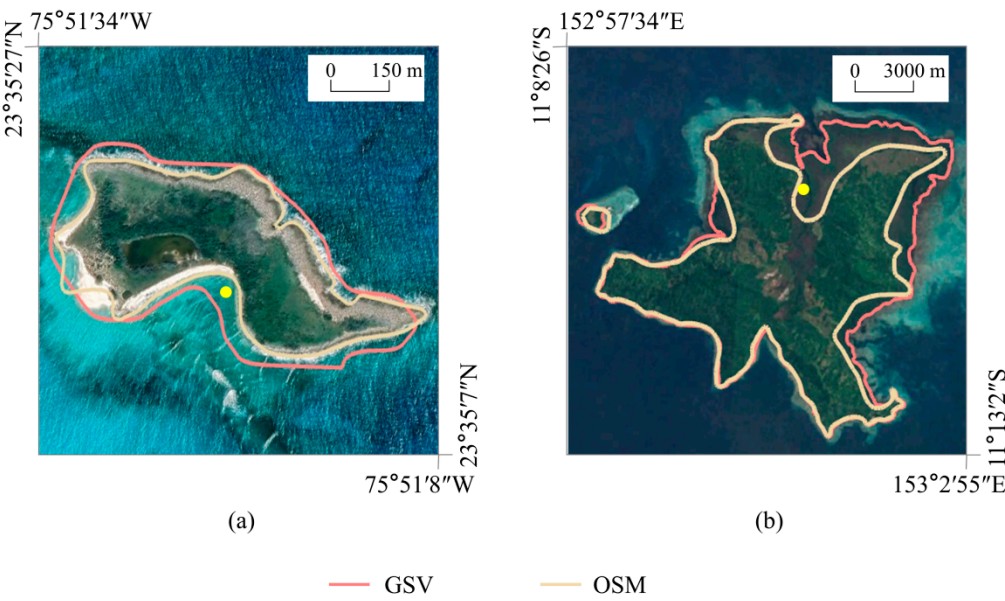

**Figure 5.** Illustrating the performances of two island datasets (GSV and OSM) by overlapping them with satellite images (**a**,**b**) in Google Earth.

### 4.2. Results of Completeness

Next, Figure 6 shows the area completeness and the count completeness, respectively, for the four study areas.

In terms of area completeness (Figure 6a), most of the values are close to 100%, indicating that the total areas are almost the same for these two island datasets. Nevertheless, the area completeness is relatively low (89%) for study area I but relatively high (110%) for study area III, respectively. This indicates that in study area I, the total areas are relatively larger for the GSV dataset, but in study area III, the total areas are relatively smaller for this dataset.

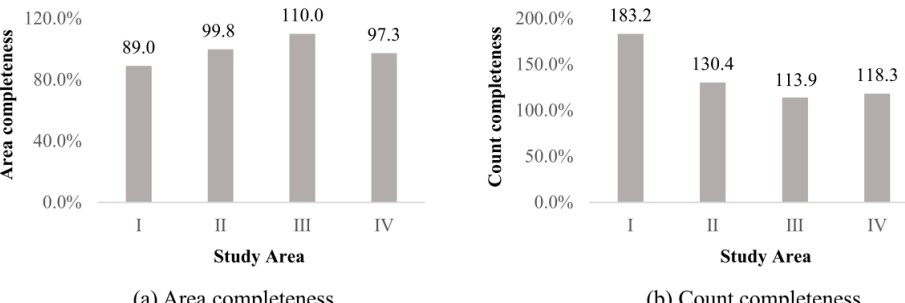

**Figure 6.** Results of the area completeness and count completeness for the four study areas (I, II, III and IV).

In terms of count completeness, the values varied dramatically from 113% to 183%. More importantly, all the values are higher than 100%, indicating that there are more islands in the OSM dataset than in the GSV dataset. Furthermore, Figure 7 also shows the number of islands in these two datasets for each study area. Unlike for Figure 6, the number was counted by taking different area intervals ($0-10^2$, $10^2-30^2$, $30^2-50^2$, $50^2-100^2$, $100^2-1000^2$, and $>1000^2$ ($m^2$)) into consideration.

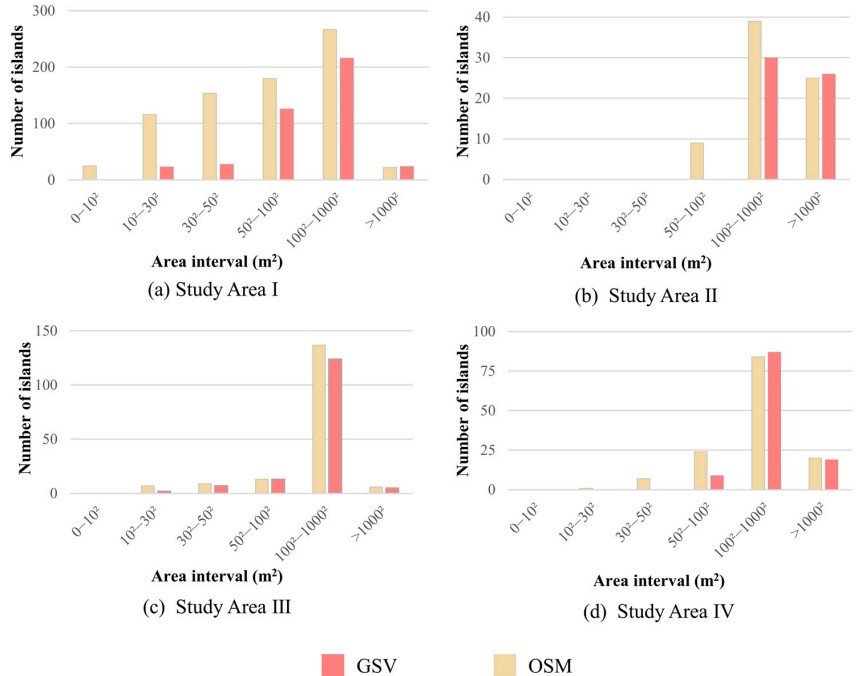

**Figure 7.** The number of islands (x-axis) in the two datasets (GSV and OSM) and for four study areas, considering different area intervals of islands (y-axis).

Figure 7 shows that the number of islands increased from the interval of $0-10^2$ to $100^2-1000^2$ ($m^2$) along with an increase in area intervals. In most cases, the number of islands is much higher in the OSM dataset than in the GSV dataset, especially for area intervals between 0 and $100^2$ ($m^2$). This indicates that there are many more small islands in the OSM dataset.

Furthermore, we investigated whether the islands in the two datasets (GSV and OSM) could also be visually interpreted from satellite images in Google Earth (Table 3). The results are reported considering different area intervals. Table 3 shows that most of the islands in each dataset can also be found in Google Earth. For instance, for study area I and an area interval of $0-10^2$ ($m^2$), 25 islands were identified in the OSM dataset, 23 of which

can be found in Google Earth. The results indicate the reliability of using these datasets for island mapping.

**Table 3.** The reliability of islands in the two datasets (GSV and OSM), in consideration of different area intervals *.

| Area Interval(m²) | I | | II | | III | | IV | |
|---|---|---|---|---|---|---|---|---|
| | **GSV** | **OSM** | **GSV** | **OSM** | **GSV** | **OSM** | **GSV** | **OSM** |
| $0–10^2$ | 0/0 | 23/25 | 0/0 | 0/0 | 0/0 | 0/0 | 0/0 | 0/0 |
| $10^2–30^2$ | 20/23 | 113/116 | 0/0 | 0/0 | 1/2 | 6/7 | 0/0 | 1/1 |
| $30^2–50^2$ | 27/28 | 151/154 | 0/0 | 0/0 | 6/7 | 9/9 | 0/0 | 7/7 |
| $50^2–100^2$ | 123/126 | 178/180 | 0/0 | 7/9 | 12/13 | 12/13 | 9/9 | 24/24 |
| $100^2–1000^2$ | 216/216 | 267/267 | 29/30 | 39/39 | 124/124 | 137/137 | 79/87 | 84/84 |
| $>1000^2$ | 24/24 | 22/22 | 26/26 | 25/25 | 5/5 | 6/6 | 19/19 | 20/20 |

* The number to the left of the "/": represents the count of islands present not only in the dataset but also visible on Google Earth; the number to the right of the "/": indicates the total number of islands identified in this dataset.

Despite this advantage, flaws may also be found. Specifically, a few islands, either in the GSV dataset (Figure 8a) or in the OSM dataset (Figure 8b), cannot be found in Google Earth, indicating errors in these two island datasets. Additionally, the number of islands in the OSM dataset is higher, probably due to two reasons. On the one hand, more islands with a relatively small area (e.g., $<100^2$ (m²)) can be identified in the OSM dataset. On the other hand, two or multiple small islands have been mapped as integrated into one in the GSV dataset (Figure 8c,d).

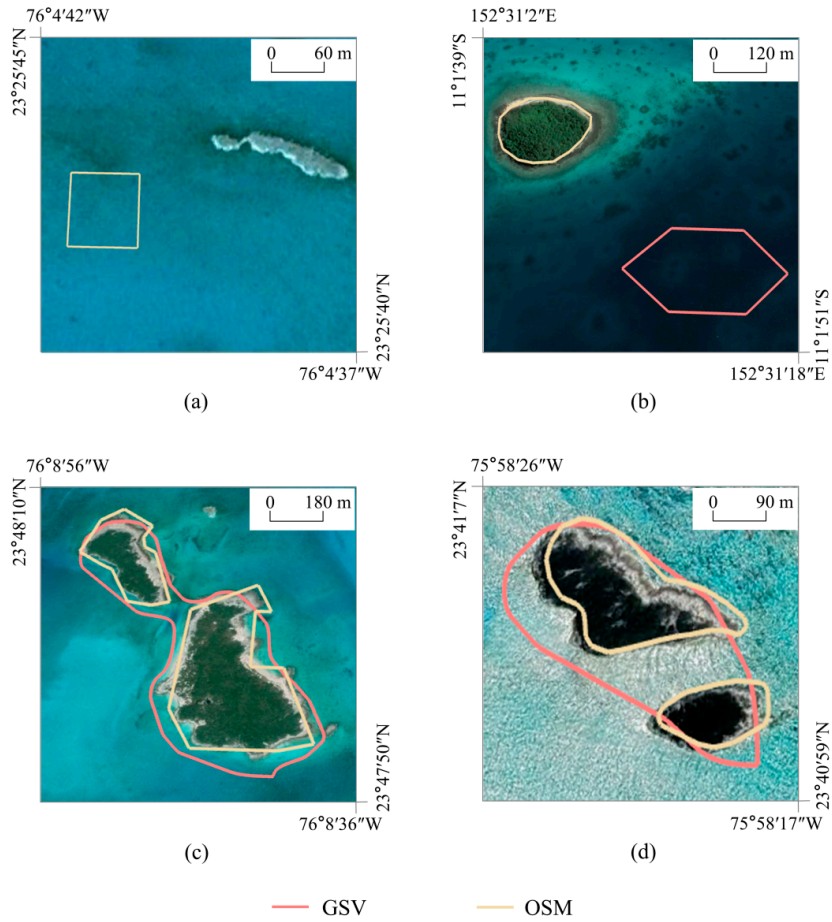

**Figure 8.** Overlapping the two island datasets with corresponding satellite images (**a–d**) in Google Earth, in order to understand the results in Table 3.

### 4.3. Results of Shape Complexity

Figure 9 further plots the results of the box-counting method. The corresponding fractal dimensions for the two island datasets (OSM and GSV) and for the four study areas (I, II, III and IV) are also provided.

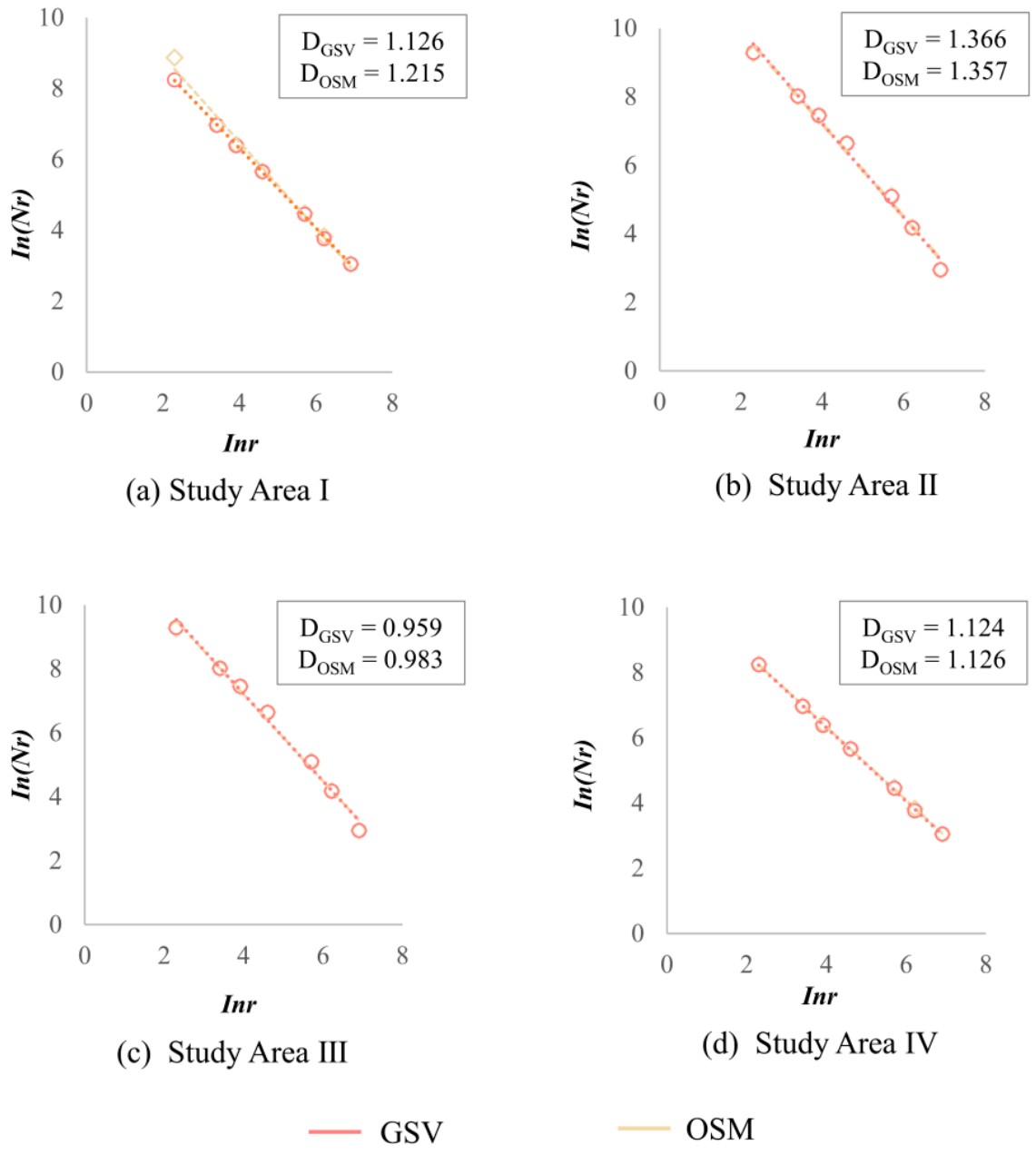

**Figure 9.** Results of the box-counting method and corresponding fractal dimensions, in terms of the two island datasets (OSM and GSV) and the four study areas (I, II, III and IV).

Figure 9 shows that the fractal dimension varies with different study areas, probably because the number of islands varies in different study areas (see Table 2). For the same study area, the fractal dimensions are almost the same for the two island datasets (GSV and OSM), somehow indicating the similarity between them. In most cases, the fractal dimension is a bit larger for the OSM dataset than for the GSV dataset. This indicates that the boundary is relatively more complex for the OSM dataset (or the island data has more details), although this is not the case for study area II. In order to further understand the results, Figure 10 shows two examples. Each island dataset in these examples is overlapped

not only with satellite images in Google Earth (Figure 10a,d) but also with regular grids (Figure 10b,c,e,f).

We can see from Figure 10 that, generally, the OSM dataset includes more details, probably because it is more precise (Figure 4). As an example, the two islands in Figure 10a can be seen on Google Earth and can also be identified in the OSM dataset. However, in the GSV dataset, only a single larger island (which includes the two relatively small ones) can be identified. Thus, relatively more grid cells (200) overlap with the OSM dataset than with the GSV dataset (195).

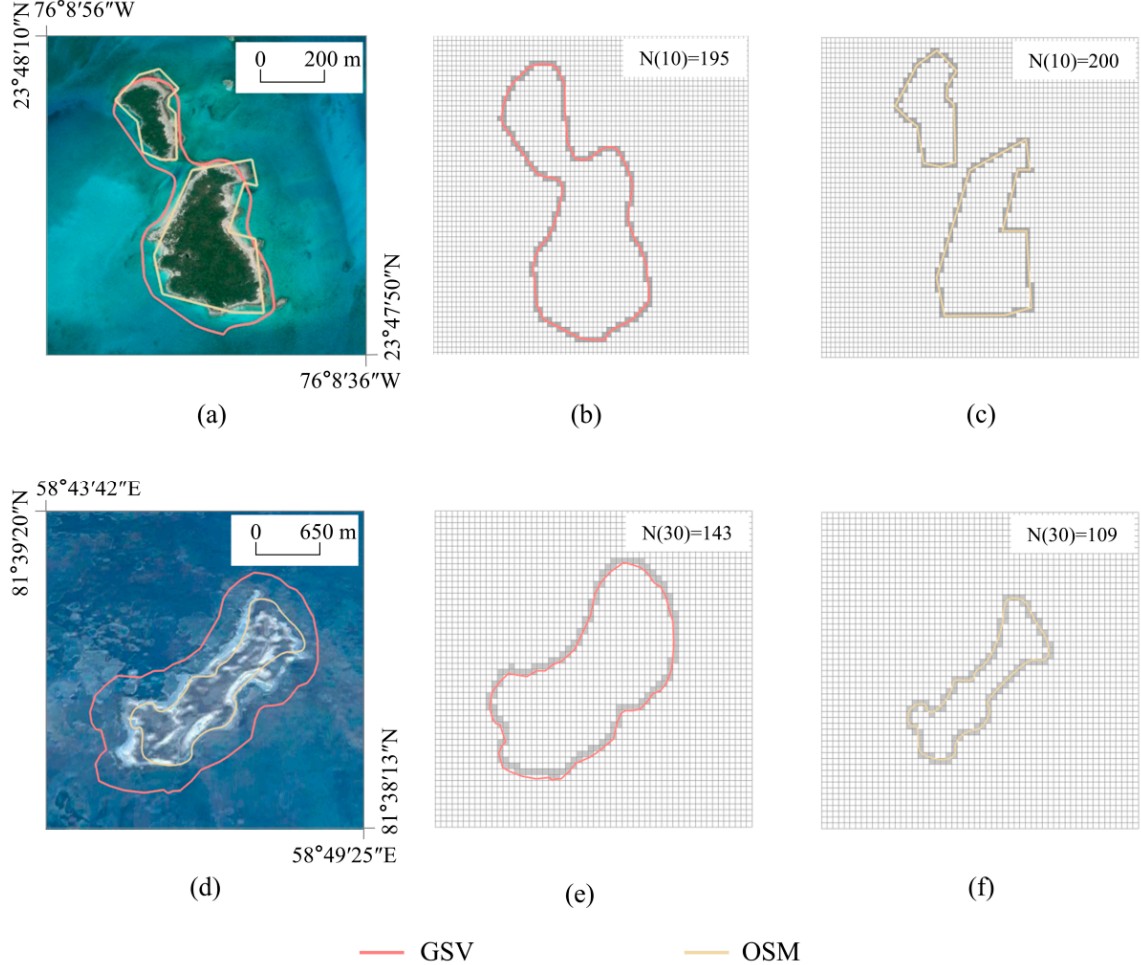

**Figure 10.** Two examples (**a**–**c** and **d**–**f**) are used to understand the results in Figure 9. For each example, both datasets are superimposed on either a Google satellite image or a regular grid.

In contrast, in Figure 10d–f, relatively more grid cells (143) overlap with the GSV dataset than with the OSM dataset (109), probably because the perimeter of the island in the GSV dataset is longer than that in the OSM dataset. However, the island in the OSM dataset still appears to be more precise when visually interpreting the satellite image on Google Earth.

## 5. Discussion

### 5.1. Comparing between GSV and OSM Datasets

This study assessed the data quality of two categories of ocean island datasets (GSV and OSM). This was achieved not only by comparing each dataset with a set of reference sampling points visually interpreted from Google Earth in terms of accuracy but also by comparing these two datasets in terms of completeness and shape complexity. Therefore, it is interesting to investigate which dataset can perform better. Our results showed that:

From an accuracy aspect, the OSM dataset performs better than the GSV dataset in terms of precision, but the GSV dataset performs better than the OSM dataset in terms of recall. However, in terms of overall accuracy (OA) and F1, the best performance for using the GSV and OSM datasets varies among the different study areas.

From the completeness aspect, the area completeness is close to 100%, indicating that the total areas of the GSV and OSM datasets are almost the same. However, the count completeness is much larger than 100%, indicating that the number of islands acquired from the OSM dataset is much more than those acquired from the GSV dataset. Moreover, we also found that more small islands (e.g., <100 × 100 m$^2$) can be acquired from the OSM dataset than from the GSV dataset.

From the shape complexity, the fractal dimension calculated based on the OSM dataset is also slightly larger than that calculated based on the GSV dataset, indicating that in most cases, the boundary of islands in the OSM dataset has relatively more details.

Therefore, we argue that the OSM dataset performs better than the GSV dataset for most of the measures (i.e., precision, completeness, and shape complexity). This is probably because the two datasets were produced based on different spatial resolutions of remote sensing data (Figure 11), that is, the GSV dataset was produced based on Landsat 7, which has a spatial resolution of 30 m [22]. On the other hand, the OSM dataset was edited by global volunteers based on Bing satellite map, which has a much higher spatial resolution (0.5 m [32]). Thus, the islands in the OSM dataset are represented with more details. Nevertheless, the GSV dataset performs better than the OSM dataset in terms of recall. Thus, the GSV dataset can still be used as a supplement, especially when the islands of a region have not been mapped well in OSM.

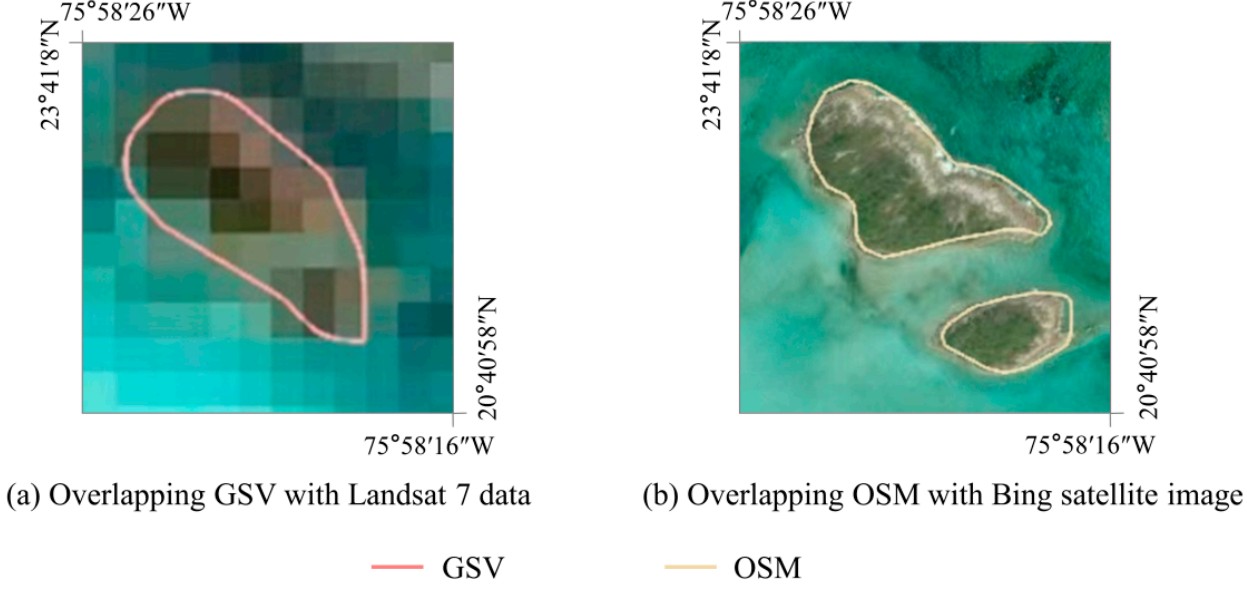

(a) Overlapping GSV with Landsat 7 data      (b) Overlapping OSM with Bing satellite image

—— GSV      —— OSM

**Figure 11.** Overlapping the islands in the GSV and OSM datasets with Landsat 7 (**a**) and Bing satellite map (**b**), respectively.

### 5.2. Applications

As both categories of island datasets (GSV and OSM) perform well in terms of accuracy (98% or above) and F1 (0.95 or above), there are several potential applications for them. First, these datasets can be used to map the spatial pattern of ocean islands not only in a region but across the globe, as both are freely available on a global scale. Moreover, this type of analysis has benefits for ship routing planning [40,41] and marine protected areas planning [42–44].

Furthermore, the OSM dataset is continuously updated by global volunteers on a minute-by-minute basis (https://wiki.openstreetmap.org/wiki/Osmupdate, accessed

on 13 February 2023). It is therefore feasible to acquire historical data of islands in OSM, which can be used to analyze the variation of islands over a long time series. This type of analysis is essential to monitor SDG-related indicators, which may be useful in achieving sustainable development of the marine environment (Virto 2018).

*5.3. Limitations*

Despite the advantages and applications of using the two island datasets, there are several limitations to this study. First, the accuracy-relevant measures were analyzed by comparing them to a set of reference sampling points that were visually interpreted from Google Earth. On the one hand, the coastline or boundary of an island may vary with different years and even different seasons. We did not consider such variation because very high-resolution satellite images (e.g., 1 m or higher) are not freely available. Although Google Earth provides high-resolution satellite images, the available years are limited and inconsistent in different study areas. On the other hand, each sampling point was only identified as 'island' (above sea surface) or 'non-island' (below sea surface). We did not divide these sampling points into more detailed classes (e.g., islands and reefs). This is because it is difficult to distinguish between the different classes through visual interpretation from Google Earth. Nevertheless, in future work, it would still be worthwhile to use other data sources to assess the data quality of these island datasets by considering more classes.

Second, we compared the GSV and OSM datasets in terms of completeness and shape complexity measures. This is because there is no corresponding reference dataset. Thus, we cannot quantitatively evaluate how complete each island dataset is or how much difference there is in shape complexity between each island dataset and a reference dataset. Moreover, only 2500 samplings were collected for each study area because visually interpreting the type of each sampling point from Google Earth is still a time-consuming and labor-intensive task. However, in future work, more sampling points should be gathered to enhance the reliability of our results.

Last but not least, in this study, only four $100 \times 100$ km$^2$ regions were chosen as the study areas. This is also because it is costly to determine the types of a large number of sampling points (10,000 in total). However, both the GSV and OSM datasets are freely available at a global scale. Therefore, in further work, it would be worthwhile to apply our analytical framework to other regions across the globe to investigate whether consistent results can be found.

## 6. Conclusions

This study assessed two categories of open island datasets (GSV and OSM) using three types of measures: accuracy, completeness, and shape complexity. Specifically, in terms of accuracy, each island dataset was compared with a set of reference sampling points that were visually interpreted with Google Earth, and four different measures, including overall accuracy (OA), precision, recall, and F1, were calculated. In terms of completeness, both area completeness and count completeness were used to compare the two island datasets. Different sizes of islands were also considered during the comparison. In terms of shape complexity, the box-counting method was employed to calculate the fractal dimension of each study area, and then the fractal dimensions of these two island datasets were compared. Four $100 \times 100$ km$^2$ regions across the globe were included as the study areas for the analysis. The results showed that:

(1) The best performance between the two island datasets (GSV and OSM) varied with different study areas in terms of OA and F1. In most cases, the OSM dataset performed better in terms of precision, but GSV performed better with respect to recall.

(2) Area completeness is close to 100%, indicating that both the GSV and OSM datasets are similar in terms of the total area of islands. However, count completeness was much higher than 100%, indicating that the OSM dataset is larger than the GSV dataset

in terms of the total number of islands. Likewise, more small islands can be acquired from the OSM dataset.

(3) In most cases, the OSM dataset has a higher value than the GSV dataset in terms of shape complexity (or fractal dimension), indicating that the OSM dataset has more details in terms of the island boundary or coastline.

We concluded that both the GSV and OSM datasets are effective, especially in terms of OA and F1, and the OSM dataset can identify more small islands and provide more details. Despite these advantages, in future work, other high-resolution remote sensing data could be used to assess the data quality of the two island datasets, especially by taking different years and seasons into consideration. Other reference datasets may also be acquired as benchmarks to carry out quantitative assessments (e.g., in terms of completeness and shape complexity). Lastly, other regions across the globe should also be involved in the analysis to verify our results.

**Author Contributions:** Conceptualization, Qi Zhou and Lihua Zhang; Formal analysis, Yijun Chen and Shenxin Zhao; Writing—original draft, Qi Zhou and Yijun Chen. All authors have read and agreed to the published version of the manuscript.

**Funding:** The project was supported by the Director Fund of the International Research Center of Big Data for Sustainable Development Goals (Grant No. CBAS2022DF010) and the National Natural Science Foundation of China (Grant No. 41771428; 4207040449).

**Data Availability Statement:** Related data are available upon reasonable request.

**Acknowledgments:** The authors thank all anonymous reviewers and the editor for their valuable comments and suggestions that have helped improve this paper substantially.

**Conflicts of Interest:** The authors declare no conflict of interest.

## Appendix A

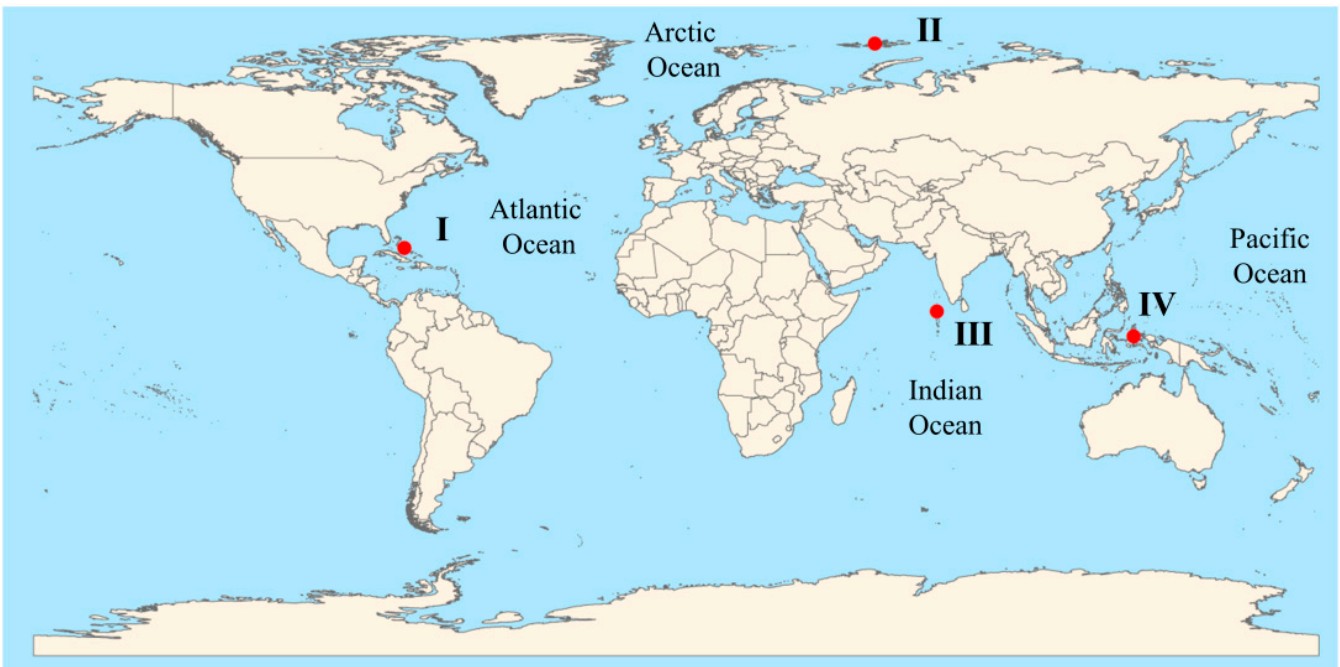

**Figure A1.** Locations of the four study areas.

## Appendix B

**Table A1.** The evaluation results of two island datasets (GSV and OSM), in terms of the accuracy measures.

| Study Area | Overall Accuracy (%) | | Precision (%) | | Recall (%) | | F1 | |
|:---:|:---:|:---:|:---:|:---:|:---:|:---:|:---:|:---:|
| | GSV | OSM | GSV | OSM | GSV | OSM | GSV | OSM |
| I | 99.64 | 99.48 | 95.24 | 98.18 | 90.91 | 81.82 | 0.93 | 0.89 |
| II | 98.40 | 98.24 | 96.65 | 96.75 | 98.54 | 97.93 | 0.98 | 0.97 |
| III | 99.96 | 99.96 | 100.00 | 100.00 | 93.33 | 93.33 | 0.97 | 0.97 |
| IV | 99.48 | 99.60 | 90.83 | 93.40 | 97.06 | 97.06 | 0.94 | 0.95 |

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
