# Peer review of "Quality Assessment of Global Ocean Island Datasets"

_ijgi, doi:10.3390/ijgi12040168_

Round 1
Reviewer 1 Report
The Manuscript reflects the very important aims of comparing the quality of data sets captured from two different approaches i.e. OSM and GSV. Although the manuscript is well written, there are some comments to the authors.
The authors have taken three parameters for data sets quality assessment which are 1) accuracy;
2) Completeness; and 3) boundary complexity. The authors are suggested to mention the justification of selecting these three parameters. In addition, it is suggested to explore the TQM “”Total Quality Management”.
The data sets of two approaches has been compared with the visualization method as author mentioned about visually interpreted with goggle earth. It really put the questions about the process applied in the data sets generation of OSM and GSV. OSM data sets matches with the visual interpretation method, however GSV is different. Author is suggested for justification.
The image of 2021 is taken as a base, however, time period of OSM data has not been mentioned. How reliable will the comparison approach of OSM data if time period of data acquisition is different?
Author Response
Reviewer 1
Reviewer1_Comment 1
The authors have taken three parameters for data sets quality assessment which are 1) accuracy; 2) Completeness; and 3) boundary complexity. The authors are suggested to mention the justification of selecting these three parameters. In addition, it is suggested to explore the TQM “Total Quality Management”.
Reply to Reviewer1_Comment 1: Thanks for this valuable comment! We have highlighted this point in the revised manuscript (see Section 3).
That is, First, both the accuracy and completeness are quality measures that defined by the ISO (International Organization for Standardization 2013), and they have been widely used to assess the quality of various types (roads, buildings, land-cover/land-use) of geo-spatial data. Second, the geometry irregularity or complexity has often been analyzed while investigating coastlines (Husain et al. 2021).
References
International Organization for Standardization. (2013). Geographic information - Data quality (ISO 19157:2013).
Husain, A., Reddy, J., Bisht, D., and Sajid, M. (2021). Fractal dimension of coastline of Australia. Scientific Reports, 11(1), 6304.
The purpose of our study is to assess and compare the data quality of two existing island datasets (GSV and OSM). However, ''Total Quality Management (TQM)'' is a system implemented by the management of an organization to achieve the satisfaction of customers or patients. TQM is held to be an innovative approach to the management of organizations. In the medical sector, TQM integrates quality orientation in all processes and procedures in health-care delivery. In our opinion, TQM highlights on 'management' rather than 'assessment'. Thus we did not consider it in the current manuscript (Alzoubi et al. 2019).
References
Alzoubi, M.M. et al. (2019). Total quality management in the health-care context: integrating the literature and directing future research. Risk Management and Healthcare Policy, 167-177.
Reviewer1_Comment 2
The data sets of two approaches has been compared with the visualization method as author mentioned about visually interpreted with goggle earth. It really put the questions about the process applied in the data sets generation of OSM and GSV. OSM data sets matches with the visual interpretation method, however GSV is different. Author is suggested for justification.
Reply to Reviewer1_Comment 2: Thanks for this valuable comment! We have explained the reason(s) in the revised manuscript (see Section 5.1). That is,
We argued that the OSM dataset performs better than the GSV dataset for most of the measures (e.g., precision, completeness and shape complexity). This is probably because the two datasets were produced based on different spatial resolutions of remote sensing data (Figure 3). That is, the GSV data was produced based on Landsat 7, which has a spatial resolution of 30m. But, the OSM data was edited by global volunteers based on Bing satellite map, which has a much higher spatial resolution (0.5m). Thus the islands in the OSM are represented with more details.
Figure 10. Overlapping the islands in the GSV and OSM datasets with Landsat 7 (a) and Bing satellite map (b), respectively.
Reviewer1_Comment 3
The image of 2021 is taken as a base, however, time period of OSM data has not been mentioned. How reliable will the comparison approach of OSM data if time period of data acquisition is different?
Reply to Reviewer1_Comment 3: Thanks for this valuable comment! We have added this point in the revised manuscript. That is (see Section 2.2),
The dataset were freely acquired in 2021 from the Plant OSM: https://planet.openstreetmap.org/.

Reviewer 2 Report
In this paper, the authors contributed the following two important work
1)Different measures (including accuracy, completeness and shape complexity) were designed 86 for assessing the data quality of island datasets.
2) Both the GSV and OSM datasets were not only assessed but also compared, in order to investigate which one can perform the best.
Therefore, it is interesting and attractive. However, it should be major revised to enhance the quality, as follows:
1) In Section 1, authors should make three sub sections, motivation, contributions and organization of the paper
2) Literature review is missing , Pl add in section 2
3) A summary table should be provided for convenience for the readers in literature review section with comparison analysis of other approaches
4) Contributions of the research paper is limited, Pl at least three contributions should be there in any journal article
5) There should be one flow chart for overall work methodology.
6) Figure 2 should be re-presented. Moreover, all the parameters should be explain clearly .
7) Eq 3,4,5,6 and 7 are not derived properly, Pl explain it in details .
8) All Figures should be enhanced at the resolution of 300 dpi.
9) Finally, the authors should double-check all formation, typos, and writing throughout the paper.
Author Response
Reviewer 2
Reviewer2_Comment 1
1) In Section 1, authors should make three sub sections, motivation, contributions and organization of the paper
Reply to Reviewer2_Comment 1: Thanks for this valuable comment! We have highlighted the motivation, objective, contribution and structure in the revised manuscript (see Introduction). That is,
Motivation: Despite of these available island datasets (GSV and OSM), to the best of our knowledge, few studies have paid attention to the data quality of these datasets; or there still is a lack of research work that assessing OSM data quality in terms of islands and/or islets.
Objective: The purpose of our study is to assess and compare the data quality of two existing island datasets (GSV and OSM).
Contribution: This study has two main contributions. 1) Different measures (including accuracy, completeness and shape complexity) were designed for assessing the data quality of island datasets. 2) Both the GSV and OSM datasets were not only assessed but also compared, in order to investigate which one can perform the best.
Structure: The paper is structured as follows: Section 2 describes the study area and data. Section 3 presents the designed measures that used for assessing and comparing GSV and OSM island datasets. Section 4 reports the results and analyses. Section 5 and Section 6 are discussions and conclusion, respectively.
We have also referred to some articles (Yu et al. 2023; Dimyati et al. 2023; Gülci et al. 2022; Chen et al. 2022; Sferlazza et al. 2022) published in the journal of IJGI. They all did not use sub-sections.
Reference
Dimyati M, Supriatna S, Nagasawa R, Pamungkas FD, Pramayuda R, (2023). A Comparison of Several UAV-Based Multispectral Imageries in Monitoring Rice Paddy (A Case Study in Paddy Fields in Tottori Prefecture, Japan). ISPRS International Journal of Geo-Information, 12(2):36.
Yu, Z., Xu, L., Chen, S., Jin, C. (2023). Research on Urban Fire Station Layout Planning Based on a Combined Model Method. ISPRS International Journal of Geo-Information, 12(3), 135.
Gülci S, Acar HH, Akay AE, Gülci N, (2022). Evaluation of Automatic Prediction of Small Horizontal Curve Attributes of Mountain Roads in GIS Environments. ISPRS International Journal of Geo-Information, 11(11):560.
Chen M, Chen Y, Wilson JP, Tan H, Chu T, (2022). Using an Eigenvector Spatial Filtering-Based Spatially Varying Coefficient Model to Analyze the Spatial Heterogeneity of COVID-19 and Its Influencing Factors in Mainland China. ISPRS International Journal of Geo-Information, 11(1):67.
Sferlazza S, Maltese A, Dardanelli G, La Mela Veca DS, (2022). Optimizing the Sampling Area across an Old-Growth Forest via UAV-Borne Laser Scanning, GNSS, and Radial Surveying. ISPRS International Journal of Geo-Information, 11(3):168.
Reviewer2_Comment 2
Literature review is missing , Pl add in section 2
Reply to Reviewer2_Comment 2: Thanks for this comment! The literature review has been added in Introduction. That is,
Remote sensing have been viewed as a potential technology to detect islands and relevant characteristics (e.g., temperature and land-use change). Dong et al. [12] developed a simple method for mapping the inundation frequency of coral reefs in the Spratly Islands in the South China Sea using time-series Landsat-8 OLI images. Immordino et al. [13] used Sentinel-2 multispectral data for mapping different types of habitats (corals, seagrasses and mangroves) of Palau Republic in the Pacific Ocean. Lyons et al. [14] presented a framework that is capable of mapping coral reef habitats from individual reefs to entire barrier reef systems and across vast ocean extents. This is also achieved by using high-resolution remote sensing data available at the global scale. Zhuang et al. [15] proposed a technical framework for automatic coral reef extraction based on an image filtering strategy and spatio-temporal similarity measurements of pixel-level Sentinel-2 image time series. Mikelsons et al. [16] developed a methodology to derive a global medium resolution (250m) land mask (or water mask) from several existing data sources. In terms of island characteristics, Král and Pavliš [17] produced the first detailed land-cover map of Socotra Island using Landsat 7 ETM+ dataset. Révillion et al. [18] have developed a land-use/land-cover product, based on remote sensing processing on high spatial resolution satellite images acquired by SPOT 5 satellite between December 2012 and July 2014. Chen et al. [19] used Landsat data for eight periods from 1984 to 2020 to explore the spatial and temporal characteristics of the land-use landscape pattern of Zhoushan Island, China. Holdaway et al. [20] analyzed the changes in land area on 221 atolls (a ring-shaped coral island or reef) in the Indian and Pacific Oceans. Leihy et al. [21] applied a spatial-temporal gap-filling method to high-resolution (~1km) land surface temperature observations for 20 Southern Ocean Islands.
Despite of these available island datasets (GSV and OSM), to the best of our knowledge, few studies have paid attention to the data quality of these datasets. The GSV dataset has only been validated using visual inspection rather than quantitative assessment [22]. Many concerns have also been paid attention to the data quality of OSM because the data was edited by global volunteers from different countries [24], different ages and education backgrounds [25]. Although extensive studies have focused on assessing OSM data quality in terms of roads [26-28], buildings [29-31], land-cover and land-uses [32-34], to the best of our knowledge, there still is a lack of research work that assessing OSM data quality in terms of islands and/or islets.
We did not add a new section for literature review because: Although existing studies have focused on detecting islands using remote sensing technology, to the best of our knowledge, few studies have paid attention to the data quality of island datasets. Moreover, we have also referred to some articles (Wu et al. 2022, Sferlazza et al. 2022, Buffa et al. 2022, Ji et al. 2022, Yu et al. 2023; Dimyati et al. 2023; Gülci et al. 2022; Chen et al. 2022; Seong et al. 2023; Kurnia et al. 2023) published in the journal of IJGI. They also did not include a section for literature review.
Reference
Wu M, Ye H, Niu Z, Huang W, Hao P, Li W, Yu B, (2022). Operation Status Comparison Monitoring of China’s Southeast Asian Industrial Parks before and after COVID-19 Using Nighttime Lights Data. ISPRS International Journal of Geo-Information, 11(2):122.
Sferlazza S, Maltese A, Dardanelli G, La Mela Veca DS, (2022). Optimizing the Sampling Area across an Old-Growth Forest via UAV-Borne Laser Scanning, GNSS, and Radial Surveying. ISPRS International Journal of Geo-Information, 11(3):168.
Buffa C, Sagan V, Brunner G, Phillips Z, (2022). Predicting Terrorism in Europe with Remote Sensing, Spatial Statistics, and Machine Learning. ISPRS International Journal of Geo-Information, 11(4):211.
Ji, H., Wang, J., Zhu, Y., Shi, C., Wang, S., Zhi, G., Meng, B. (2022). Spatial Distribution of Urban Parks’ Effect on Air Pollution-Related Health and the Associated Factors in Beijing City. ISPRS International Journal of Geo-Information, 11(12), 616.
Dimyati M, Supriatna S, Nagasawa R, Pamungkas FD, Pramayuda R, (2023). A Comparison of Several UAV-Based Multispectral Imageries in Monitoring Rice Paddy (A Case Study in Paddy Fields in Tottori Prefecture, Japan). ISPRS International Journal of Geo-Information, 12(2):36.
Yu, Z., Xu, L., Chen, S., Jin, C. (2023). Research on Urban Fire Station Layout Planning Based on a Combined Model Method. ISPRS International Journal of Geo-Information, 12(3), 135.
Gülci S, Acar HH, Akay AE, Gülci N, (2022). Evaluation of Automatic Prediction of Small Horizontal Curve Attributes of Mountain Roads in GIS Environments. ISPRS International Journal of Geo-Information, 11(11):560.
Chen M, Chen Y, Wilson JP, Tan H, Chu T, (2022). Using an Eigenvector Spatial Filtering-Based Spatially Varying Coefficient Model to Analyze the Spatial Heterogeneity of COVID-19 and Its Influencing Factors in Mainland China. ISPRS International Journal of Geo-Information, 11(1):67.
Seong J, Kim Y, Goh H, Kim H, Stanescu A, (2023). Measuring Traffic Congestion with Novel Metrics: A Case Study of Six U.S. Metropolitan Areas. ISPRS International Journal of Geo-Information, 12(3):130.
Kurnia, A. A., Rustiadi, E., Fauzi, A., Pravitasari, A. E., Ženka, J. (2023). Probing Regional Disparities and Their Characteristics in a Suburb of a Global South Megacity: The Case of Bekasi Regency, Jakarta Metropolitan Region. ISPRS International Journal of Geo-Information, 12(2), 32.
Reviewer2_Comment 3
3) A summary table should be provided for convenience for the readers in literature review section with comparison analysis of other approaches
Reply to Reviewer2_Comment 3: Thanks for this comment! Although extensive studies have focused on detecting islands using remote sensing technology, to the best of our knowledge, few studies have paid attention to the data quality of island datasets. Thus we did not give out a summary table to report how to detect islands, which is out of our scope.
Reviewer2_Comment 4
4) Contributions of the research paper is limited, Pl at least three contributions should be there in any journal article
Reply to Reviewer2_Comment 4: Thanks for this comment! The main contribution of our study is to assess and compare the data quality of two existing island datasets (GSV and OSM). This is because to the best of our knowledge, few studies have paid attention to the data quality of island datasets.
We think the contribution of a research paper depends on the quality rather than on the quantity. For instances, We list some papers that published in the journal of IJGI that only have 1-2 listed contributions also.
Reference
Seong J, Kim Y, Goh H, Kim H, Stanescu A, (2023). Measuring Traffic Congestion with Novel Metrics: A Case Study of Six U.S. Metropolitan Areas. ISPRS International Journal of Geo-Information, 12(3):130.
Kurnia, A. A., Rustiadi, E., Fauzi, A., Pravitasari, A. E., Ženka, J. (2023). Probing Regional Disparities and Their Characteristics in a Suburb of a Global South Megacity: The Case of Bekasi Regency, Jakarta Metropolitan Region. ISPRS International Journal of Geo-Information, 12(2), 32.
Buffa C, Sagan V, Brunner G, Phillips Z, (2022). Predicting Terrorism in Europe with Remote Sensing, Spatial Statistics, and Machine Learning. ISPRS International Journal of Geo-Information, 11(4):211.
Dimyati M, Supriatna S, Nagasawa R, Pamungkas FD, Pramayuda R, (2023). A Comparison of Several UAV-Based Multispectral Imageries in Monitoring Rice Paddy (A Case Study in Paddy Fields in Tottori Prefecture, Japan). ISPRS International Journal of Geo-Information, 12(2):36.
Chen M, Chen Y, Wilson JP, Tan H, Chu T, (2022). Using an Eigenvector Spatial Filtering-Based Spatially Varying Coefficient Model to Analyze the Spatial Heterogeneity of COVID-19 and Its Influencing Factors in Mainland China. ISPRS International Journal of Geo-Information, 11(1):67.
Wu M, Ye H, Niu Z, Huang W, Hao P, Li W, Yu B, (2022). Operation Status Comparison Monitoring of China’s Southeast Asian Industrial Parks before and after COVID-19 Using Nighttime Lights Data. ISPRS International Journal of Geo-Information, 11(2):122.
Sferlazza S, Maltese A, Dardanelli G, La Mela Veca DS, (2022). Optimizing the Sampling Area across an Old-Growth Forest via UAV-Borne Laser Scanning, GNSS, and Radial Surveying. ISPRS International Journal of Geo-Information, 11(3):168.
Ji, H., Wang, J., Zhu, Y., Shi, C., Wang, S., Zhi, G., Meng, B. (2022). Spatial Distribution of Urban Parks’ Effect on Air Pollution-Related Health and the Associated Factors in Beijing City. ISPRS International Journal of Geo-Information, 11(12), 616.
Reviewer2_Comment 5
5) There should be one flow chart for overall work methodology.
Reply to Reviewer2_Comment 5: Thanks for this valuable comment! A flow chart has been added. That is (see Section 3).
Figure 2. The workflow of our evaluation method.
Reviewer2_Comment 6
6) Figure 2 should be re-presented. Moreover, all the parameters should be explain clearly .
Reply to Reviewer2_Comment 6: Thanks for this valuable comment! The Figure 2 has been re-presented and all the parameters have been explained in the revised manuscript. That is,
Figure 3. The principle of the box counting method.
Here, r denotes the size of grid cell; and Nr denotes the number of grid cells that intersecting with a line or boundary.
Reviewer2_Comment 7
7) Eq 3,4,5,6 and 7 are not derived properly, Pl explain it in details .
Reply to Reviewer2_Comment 5: Thanks for this valuable comment! All the parameters in these Eqs have been explained.
The predicted type was compared with the corresponding reference type for all the sampling points, in terms of four different measures (i.e., overall accuracy (OA), precision, recall and F1).
OA=(TP+TN)/(TP+FP+TN+FN)×100% (1)
Precision=FP/(FP+TN)×100% (2)
Recall=TP/(TP+FN)×100% (3)
F1= (2×Precision×Recall)/(Precision+Recall) (4)
Where, TP denotes the number of sampling points that were identified as 'island' in both the open dataset and Google Earth; TN denotes the number of sampling points that were identified as 'non-island' in both the open dataset and Google Earth; FP denotes the number of sampling points that were classified as 'island' in the open dataset but they were interpreted as 'non-island' in Google Earth; and FN denotes the number of sampling points that were classified as 'non-island' in the open dataset but they were interpreted as 'island' in Google Earth.
Two measures (area completeness and count completeness) were employed to assess the completeness [29]. The two measures (called C_area and C_count) are described as follows.
C_area = A_OSM/A_GSV × 100% (5)
C_count=N_OSM/N_GSV × 100% (6)
Where, A_GSV and A_OSM denote the total areas of islands in the GSV and OSM datasets, respectively. N_OSM and N_GSV denote the total number of islands in the GSV and OSM datasets, respectively.
The islands of each dataset (e.g., GSV or OSM), originally represented by polygons, were converted into lines (or boundaries). Then, the lines or boundaries in each dataset and each study area (i.e., I, II, III or IV) was respectively overlapped with regular girds of different sizes (i.e., 10, 30, 50, 100, 300, 500 and 1000m, Figure 3).
For each grid cell, not only the size of grid cell (r) was recorded, but also the number of grid cells that intersecting with a line or boundary (Nr) was calculated. After that, the natural logarithms of each pair of r and Nr were calculated. That means each pair of r and Nr was converted into In(r) and In(Nr), respectively.
Lastly, a linear function was used to fit multiple pairs of In(r) and In(Nr) of different grid sizes, where C denotes a constant; and D denotes the fractal dimension.
In(Nr) = -Dln(r) + lnC (7)
where C denotes a constant; and D denotes the fractal dimension.
Reviewer2_Comment 8
8) All Figures should be enhanced at the resolution of 300 dpi.
Reply to Reviewer2_Comment 8: Thanks for this valuable comment! All the figures have been enhanced at the resolution of 600 dpi.

Reviewer 3 Report
Following changes have been made before consideration for
publication
1. In the Introduction Section, motivation, objective, and
contribution need to be presented clearly.
2. Please explain, why you have chosen these two areas, and give specific reason.
3. What the different parameters are taken into consideration, needs to be specified clearly.
4. In methodology section, nothing is explained clearly. You have to specify, which and how you have done this work with proper example, model and explanation.
5. Performance measures should be considereation into an another
section rather than methodology section.
6. In conclusion section, drawback and future scope, need to be highlighted.
Author Response
Reviewer 3
Reviewer3_Comment 1
1. In the Introduction Section, motivation, objective, and contribution need to be presented clearly.
Reply to Reviewer3_Comment 1: Thanks for this valuable comment! We have highlighted the motivation, objective, contribution and structure in the revised manuscript (see Introduction). That is,
Motivation: Despite of these available island datasets (GSV and OSM), to the best of our knowledge, few studies have paid attention to the data quality of these datasets; or there still is a lack of research work that assessing OSM data quality in terms of islands and/or islets.
Objective: The purpose of our study is to assess and compare the data quality of two existing island datasets (GSV and OSM).
Contribution: This study has two main contributions. 1) Different measures (including accuracy, completeness and shape complexity) were designed for assessing the data quality of island datasets. 2) Both the GSV and OSM datasets were not only assessed but also compared, in order to investigate which one can perform the best.
Structure: The paper is structured as follows: Section 2 describes the study area and data. Section 3 presents the designed measures that used for assessing and comparing GSV and OSM island datasets. Section 4 reports the results and analyses. Section 5 and Section 6 are discussions and conclusion, respectively.
Reviewer3_Comment 2
2. Please explain, why you have chosen these two areas, and give specific reason.
Reply to Reviewer3_Comment 2: Thanks for this valuable comment! We have highlighted this point in the revised manuscript (see Section 2). That is,
Four different study areas were chosen for the analysis. This is because: First of all, they are located in different geographical regions of the world (i.e., the Atlantic Ocean, the Arctic Ocean, the Indian Ocean and the Pacific Ocean). Second, the size and pattern of islands varies with different regions (Table 1). Third and more important, four different study areas were involved in order to reduce the bias for the analysis.
Reviewer3_Comment 3
3. What the different parameters are taken into consideration, needs to be specified clearly.
Reply to Reviewer3_Comment 1: Thanks for this valuable comment! We have highlighted this point in the revised manuscript (see Section 3).
That is, First, both the accuracy and completeness are quality measures that defined by the ISO (International Organization for Standardization 2013), and they have been widely used to assess the quality of various types (roads, buildings, land-cover/land-use) of geo-spatial data. Second, the geometry irregularity or complexity has often been analyzed while investigating coastlines (Husain et al. 2021).
References
International Organization for Standardization. (2013). Geographic information - Data quality (ISO 19157:2013).
Husain, A., Reddy, J., Bisht, D., and Sajid, M. (2021). Fractal dimension of coastline of Australia. Scientific Reports, 11(1), 6304.
Reviewer3_Comment 4
4. In methodology section, nothing is explained clearly. You have to specify, which and how you have done this work with proper example, model and explanation.
Reply to Reviewer3 Comment 4: Thanks for this valuable comment! The methodology section has been revised (see Section 3).
First of all, the purpose of our study is to assess and compare the data quality of two existing island datasets (GSV and OSM).
Second, we added sentences to explain why to choose the three parameters for the analysis. First of all, both the accuracy and completeness are quality measures that defined by the ISO (International Organization for Standardization 2013), and they have been widely used to assess the quality of various types (roads, buildings, land-cover/land-use) of geo-spatial data. Moreover, the geometry irregularity or complexity has often been analyzed while investigating coastlines (Husain et al. 2021).
Third, a workflow has also been added. That is,
Figure 2. The workflow of our evaluation method.
Reviewer3_Comment 5
5. Performance measures should be considereation into an another section rather than methodology section.
Reply to Reviewer3_Comment 5: Thanks for this comment! The main contribution of our study is to assess and compare the data quality of two existing island datasets (GSV and OSM). This is because to the best of our knowledge, few studies have paid attention to the data quality of island datasets. Therefore, the methodology section presents various (quality) measures for the assessment of island datasets.
Reviewer3_Comment 6
6. In conclusion section, drawback and future scope, need to be highlighted.
Reply to Reviewer3_Comment 6: Thanks for this comment! The drawback and future scope have been highlighted in both Discussion and Conclusion.
Discussion:
First, on one hand, the coastline or boundary of an island may vary with different years and seasons. We did not consider such variation because very high-resolution satellite images (e.g., 1m or higher) are not freely available. On the other hand, each sampling point was only identified as either an 'island' (above sea surface) or 'non-island' (below sea surface). We did not divide these sampling points into more detailed classes (e.g., islands and reefs).
Second, this is due to a lack of corresponding reference data. Thus, we cannot quantitatively evaluate how complete each island data is or how much difference there is in shape complexity between each island data and a reference data. Furthermore, only 2,500 samplings were picked up for each study area because it is still a time-consuming and labor-intensive task to visually interpret the type of each sampling point from Google Earth.
Last but not least, in this study, only four 100×100 km2 regions were chosen as study areas. This is also because it is costly to determine the types of a large number of sampling points (10,000 in total).
Conclusion:
In future work, first, other high-resolution remote sensing data may be used to assess the data quality of the two island datasets, particularly by taking into consideration different years and seasons. Second, other reference datasets may be acquired as benchmarks to carry out a quantitative assessment (e.g., in terms of completeness and shape complexity). Lastly, other regions across the globe should also be included in the analysis to verify our results.
Despite of these advantages, in the future work, first, other high-resolution remote sensing data may be used to assess the data quality of the two island datasets, especially by taking different years and different seasons into consideration. Second, other reference dataset(s) may be acquired as benchmark(s) to carry out a quantitative assessment (e.g., in terms of the completeness and shape complexity). Last but not least, other regions across the globe should also be involved for the analysis, in order to verify our results.

Round 2
Reviewer 2 Report
Authors are addressed all the queries related to the present manuscript. Now , it may considered for the publications in this journal.
Pl do the following minor correction on the present manuscript.
1. Pl include one contribution subsection in section 1 and organizations subsection at the final end of section 1.
2. Pl remove first line 3.1 Accuracy which is placed two times
Author Response
Reviewer 2
Reviewer2_Comment 1
Pl include one contribution subsection in section 1 and organizations subsection at the final end of section 1.
Reply to Reviewer2_Comment 1: Thanks for this valuable comment! We have included several sub-sections in Introduction in the revised manuscript. That is,
1.1 Related works
1.2 Aim and contributions
1.3 Organization
Reviewer2_Comment 2
Pl remove first line 3.1 Accuracy which is placed two times
Reply to Reviewer2_Comment 2: Thanks for this valuable comment! We have revised the mistake in the manuscript (see Section 3.1).
